# Controversies in the Front-Line Treatment of Systemic Peripheral T Cell Lymphomas

**DOI:** 10.3390/cancers15010220

**Published:** 2022-12-30

**Authors:** Marc Sorigue, Outi Kuittinen

**Affiliations:** 1Department of Hematology, ICO-IJC-Hospital Germans Trias i Pujol, LUMN, UAB, 08916 Badalona, Spain; 2Institute of Clinical Medicine, Faculty of Health Medicine, University of Eastern Finland, 70211 Kuopio, Finland; 3Medical Research Centre and Cancer and Translational Research Unit, Oulu University Hospital and University of Oulu, 90220 Oulu, Finland; 4Department of Oncology, Kuopio University Hospital, 70210 Kuopio, Finland

**Keywords:** T cell lymphoma, stem cell transplant, etoposide, CHOP, brentuximab

## Abstract

**Simple Summary:**

Systemic peripheral T cell lymphomas (PTCL) are a rare and clinically and biologically heterogeneous group of disorders with scarce and generally low-quality evidence guiding their management. In this manuscript, we tackle the current controversies in the front-line treatment of systemic PTCL and give our thoughts vis-a-vis potential developments to come in the next few years.

**Abstract:**

Systemic peripheral T cell lymphomas (PTCL) are a rare and clinically and biologically heterogeneous group of disorders with scarce and generally low-quality evidence guiding their management. In this manuscript, we tackle the current controversies in the front-line treatment of systemic PTCL including (1) whether CNS prophylaxis should be administered; (2) whether CHOEP should be preferred over CHOP; (3) what role brentuximab vedotin should have; (4) whether stem cell transplant (SCT) consolidation should be used and whether autologous or allogeneic; (5) how should molecular subtypes (including DUSP22 or TP63-rearranged ALCL or GATA3 or TBX21 PTCL, NOS) impact therapeutic decisions; and (6) whether there is a role for targeted agents beyond brentuximab vedotin.

## 1. Introduction

Systemic peripheral T cell lymphomas (PTCL) are a rare and clinically and biologically heterogeneous group of disorders including, most commonly, PTCL, not otherwise specified (NOS), T follicular helper (TFH) lymphomas including AITL, and anaplastic large T cell lymphoma (ALCL). They are, however, characterized by poor outcomes and, precisely due to their rarity, there is a general lack of high-quality evidence grounding the therapeutic recommendations [1,2].

The front-line treatment of PTCL, NOS, TFH lymphomas, and ALCL has long been structured around anthracycline-based chemotherapy, most commonly the CHOP (cyclophosphamide, doxorubicin, vincristine, and prednisone) regimen, although this regimen became the standard of care based on trials that included mostly patients with diffuse large B cell lymphoma (DLBCL). Clinical courses with this strategy are characterized by high response rates but frequent relapses and poor outcomes, with 5-yr overall survival (OS) ~30–40% [3,4]. After relapse, response rates to salvage chemotherapy or single agent strategies are poor and survival is dismal, particularly for patients who are not eligible for—or cannot undergo due to progressive disease—stem cell transplantation (SCT) [5,6]. Reviews delving in more depth into the standard front-line treatment of PTCL have been published recently [7,8] including in this Special Issue of Cancers, and the current manuscript focuses specifically on the controversial aspects.

## 2. Current Controversies in the Treatment of Systemic Peripheral T Cell Lymphomas

### 2.1. Should CNS Prophylaxis Be Administered to Patients with Systemic PTCL?

Assessment of the suitability of CNS prophylaxis requires (1) accurate knowledge of the incidence of CNS relapse in general and patient specific risk factors, and (2) proof of decreased incidence of CNS relapse when prophylaxis is given.

Based on the low overall incidence of T cell lymphoma and the rarity of CNS relapse, estimates of the risk of CNS relapse in T cell lymphoma are imprecise. In the largest study including more than 1000 patients with PTCL, almost none of which received CNS prophylaxis, only 1.5% developed CNS relapse [9]. This is in line with the results of the MD Anderson Cancer single center data review of PTCL (which also included extranodal NK-cell lymphoma), which reported a 1-year and 5-year cumulative incidence of CNS relapse of 1.5% and 2.1%, respectively [10]. However, higher relapse rates have been reported in the Swedish registry (~4% of patients, incidence of 5.5% at 2 years [11]), and in three smaller single-center reports, which reported crude incidences of CNS relapse of ~6% [12], ~8% [13], and ~9% [14], respectively. These differences are difficult to explain and do not appear to be clearly due to between-study differences in histotypes (with PTCL-NOS, ALCL, and AITL being the most common in all studies and contributing to the majority of CNS relapse events), diagnosis of CNS relapse (CSF cytology or biopsy but also based on symptoms and imaging), treatment regimens (CHOP/CHOEP in a majority of patients), or the use of CNS prophylaxis (rarely given). Importantly, however, outside of ATLL, which is most often excluded from these retrospective studies due to its well-known greater risk of CNS relapse [12,15], and despite some discordant results, the totality of the evidence seems to suggest that ALK + ALCL is associated with a greater risk of CNS relapse (ranging from 5 up to 14%) than the other subtypes [9,10,11,12,14]. Regarding whether CNS relapses occur in the CNS in isolation or at the time of systemic relapse, very few studies have tackled this question; two studies suggest that isolated CNS relapse occurs in ~50% of patients [11,12], while another suggests only a minority of patients relapse in the CNS alone (10%, 2/20 [14]).

More agreement can be found on relapses occurring very early in the course of the disease (although acknowledging that for patients relapsing after a number of lines of chemotherapy and without further treatment options, a definitive diagnosis of CNS relapse may not be sought) and on the risk factors for CNS relapse. Most studies concur on the notion that extranodal involvement (particularly >1 site) or the involvement of a particular extranodal site (e.g., paranasal sinuses [14], gastrointestinal tract [11]) at diagnosis is a major risk factor for CNS relapse [9,10,12]. Some authors have attempted to devise a score to determine the risk of CNS relapse based on readily available data, similar to the available scores for aggressive B cell malignancies (CNS IPI), but these efforts have thus far not been externally validated [14].

The negative prognostic impact of CNS relapse has not been clearly established [9,11,14], although this is likely due to the limited statistical power and the already very poor prognosis of relapsing PTCL, regardless of CNS involvement.

Obtaining the proof of efficacy of CNS prophylaxis is likely to be even harder than determining the risk of CNS relapse because prospective data will not be available and retrospective studies are plagued by numerous confounders that prevent assessing the value of any therapeutic interventions, particularly one directed at preventing the occurrence of a low-risk event. In any case, and with the caveat of very limited statistical power for this comparison, most studies have found no protective effect of CNS prophylaxis on CNS relapse [11,12]. Furthermore, the issue of the efficacy of CNS prophylaxis is currently a hot topic in DLBCL and aggressive B cell malignancies, where very large datasets have been leveraged, and it was uniformly found that CNS prophylaxis was not effective in most patients—even if a benefit could not be excluded for specific subsets [16,17,18].

To conclude, should CNS prophylaxis be administered to patients with systemic PTCL? Despite unclear estimates of the risk of CNS relapse, some patients with PTCL are likely to be at high risk of CNS relapse. Unfortunately, the current evidence cannot firmly establish which prophylactic strategy to try to deploy. However, proof of efficacy of CNS relapse prophylaxis is lacking and data from other lymphomas suggest the current strategies (including IV methotrexate and intrathecal therapy) do not work. Therefore, we believe that at present, CNS prophylaxis should not be offered to most patients with PTCL.

### 2.2. Should CHOEP Be Preferred over CHOP?

CHOP has been the basis of the treatment of PTCL for decades. Given the poor outcomes of patients with PTCL, it stands to reason that improvements on this standard should be sought. Among these strategies to improve upon CHOP is the addition of etoposide (CHOEP), which was first suggested to improve the outcomes in PTCL in a retrospective analysis of multiple trials including not only phase III but also phase II and dose-finding studies of the German high-grade lymphoma study group (DSHNHL). In 320 patients, the authors found nominally better event-free survival (EFS) for etoposide-containing regimens than for CHOP, with statistically significant results for patients 60 or younger with normal LDH (3-yr EFS estimates 75% vs. 51%) [19]. Above the age of 60, the additional toxicity and treatment delays incurred with CHOEP seem to offset the higher efficacy. When analyzing histotypes, a statistically significant improvement in EFS was only found for patients ALK + ALCL (3-yr EFS estimates: 91% vs. 57%), with a trend for the rest of the histotypes. OS was not improved for any subgroup of patients. For proper interpretation of this analysis, the large number of comparisons without a pre-specified hypothesis and the heterogeneous treatment strategies (including multiple trials testing different regimens and drug doses) should be kept in mind.

Numerous studies have subsequently compared CHOP with CHOEP (or other etoposide-containing regimens). Most of these studies indicated some benefit for CHOEP over CHOP, but always with caveats, either limited to a histological subtype (e.g., ALK + ALCL), to young patients, or to progression-free survival (PFS)—not OS—and often with indirect evidence of patients in the CHOEP group being fitter (and having poorer prognostic features), thus limiting the conclusions to be drawn.

For instance, in one of the largest studies, Janikova et al. [20] found that improvements in PFS and OS for patients treated with CHOEP over CHOP were far more likely to aim for autologous SCT (57% vs. 41% of patients) despite a similar age at diagnosis; similarly, in the Dutch registry, a large unadjusted OS difference in favor of CHOEP vanished when adjustments for age, histotype, IPI, and subsequent use of autologous SCT were conducted [3]; finally, in an analysis of the international T-cell project including 199 ALK − ALCL treated with anthracycline (*n* = 168) or anthracycline/etoposide-based regimens (*n* = 31), a borderline difference favoring the latter was found for OS but not PFS [21]. Based on the results and the shape of the curves, both the notion that CHOEP is truly more effective but also that CHOEP-treated patients have a lower mortality risk beyond the disease itself, remain possible—and indeed, both may be true to some extent.

Regarding age and histological subtype, the Swedish registry confirmed the results of the DSHNHL study that showed that the benefit of adding etoposide seems limited to patients younger than 60 [22]; and a Danish and Swedish registry study as well as a recent Dutch registry study confirmed a PFS and OS benefit of a substantial effect size in ALK + ALCL (HR for death with CHOP 2.6 and 6.3 in the two studies, respectively) [3,23].

There is limited evidence of further treatment intensification beyond CHOEP with higher drug doses (e.g., DA-EPOCH, megaCHOEP), but overall, these mostly small trials have not suggested improved outcomes and have confirmed increased treatment toxicity (reviewed in [24]).

To conclude, should CHOEP be preferred over CHOP? CHOEP is more toxic than CHOP and the true potential benefits in PTCL other than ALK + ALCL seem small and limited to younger patients. It has been well-established that positive results in retrospective and observational analyses of oncological treatment strategies often fail to replicate—or do so with a far more modest effect size—in clinical trials [25,26,27]. However, with the current data, the 2015 ESMO guidelines favor CHOEP over CHOP [28]. We rather recommend an in-depth discussion with the patient including the limited potential benefit and the limitations in the available evidence. While current recommendations often use age cutoffs (mostly 60 years), it is likely that fitness/geriatric assessments can better pick out the patients likely to benefit from CHOEP over CHOP. Future studies will have to settle this issue. Conversely, in patients with ALK + ALCL, the consistent large effect size seen in the available publications, not only in PFS but also in OS, make a stronger case for recommending CHOEP in younger patients—pending the results from randomized trials.

### 2.3. What Is the Role of Brentuximab Vedotin in the Front-Line?

Brentuximab vedotin (BV) is an anti-CD30-monometil-auristatin A immunoconjugate. In early phase trials, it showed activity in relapsed/refractory and treatment-naïve T cell lymphoma and ostensibly better survival than expected with other available strategies in non-comparative studies [29,30]. Thus, its combination with anthracycline-based strategies for the front-line treatment was of interest, particularly in ALCL, a disorder characterized by the uniform expression of CD30.

The phase III randomized trial ECHELON-2, comparing CHOP with BV-CHP, was initially published in 2019 and an update was more recently published, with 5 years of median follow-up [31,32]. It included 452 patients, 70% ALCL (48% ALK− and 22% ALK+, the latter all with IPI ≥ 2, per the inclusion criteria), with the primary goal of showing an improvement in PFS. This was achieved with 5-year PFS probabilities of 51% vs. 43% (HR 0.7, 95%CI 0.53–0.91). The analysis based on histological subtype suggested that the benefit was restricted to ALCL (HR 0.55, 95%CI 0.39–0.79), although the study was not powered to detect differences in other subtypes. With longer follow-up, this PFS benefit translated into an OS benefit for patients with ALCL, although the use of BV after relapse in the control arm may be lower than what would be expected [32].

To conclude, what is the role of brentuximab vedotin in the front-line treatment of PTCL? ECHELON-2 convincingly demonstrates that patients with ALK − ALCL or ALK + ALCL with IPI ≥2 should receive BV-CHP as front-line induction. Conversely, we do not favor the use of BV for ALK + ALCL with IPI 0–2, which were not included in ECHELON-2, precisely on account of the very good outcomes with CHOP and CHOEP. We similarly do not favor BV-CHP for non-ALCL PTCL, for which little benefit has been found. However, we acknowledge that this is a personal assessment, given that ECHELON-2 was not powered to evaluate BV-CHP in patient subgroups other than ALCL. Additional survival analyses based on CD30 expression would also help establish the role of BV-CHP in other PTCL subtypes. Indeed, for this latter group, it is unclear whether CHOEP, which many would argue is the standard over CHOP, or BV-CHP, extrapolating from all other ALCL patient subgroups, is better.

### 2.4. When Should Stem Cell Transplant Be Considered in the Front-Line? Should Autologous or Allogeneic SCT Be Preferred?

Much like chemotherapy intensification with etoposide or the addition of novel agents such as BV, another strategy aiming to increase the cure rates in PTCL is consolidation with SCT. The potential benefit of consolidation with autologous SCT is built on single arm data, most notably the prospective Nordic NLG-T-01 trial [33] and the German trial by Reimer et al. [34], both of which reported ostensibly better outcomes (5-yr PFS and OS estimates of 44% and 51%, respectively, in the former and 3-yr PFS and OS estimates of ~30% and 48%, respectively, in the latter) than historical controls with no consolidation at the time.

Subsequently, a number of retrospective studies have indeed confirmed the benefit of autologous SCT over no consolidation, particularly for patients in first complete remission (CR1). Table 1 shows the results of selected, recent, studies comparing autologous SCT with other strategies. An exhaustive review of autologous SCT is outside the scope of this manuscript and a recent review including all studies was recently published [35]. Some of the most informative studies in this area are three very recent studies, alongside the already classical Swedish registry report [22]: (1) An analysis of SCT use in ECHELON-2 patients in CR after induction (decision to transplant was not decided by the study protocol but rather was based on physician’s choice) found that PFS was prolonged with the use of SCT in both study arms [36]; (2) the Dutch registry, which found an improvement in both OS and PFS for patients with PTCL (5-yr OS estimates of 78% vs. 45%) in a study with careful statistical adjustment, given that patients who receive SCT are frequently different than those who do not [3]; and (3) a comparison of post-induction strategies in patients with PTCL other than ALK+ in CR1 by the Spanish GELTAMO group (5-yr PFS: 63% vs. 48%) [37]. Additionally, the AITL sub-study of the prospective T cell project found a benefit of autologous SCT for this specific subset of patients [38].

However, the prospective COMPLETE registry [39], two large retrospective multicenter studies, one from Europe [40] and one from the US [41] as well as the Czech registry [20] have found no survival benefit for autologous SCT consolidation.

Even if the evidence was mostly favorable, substantial caution would be warranted because confounders and bias by indication can still lead to positive results in retrospective analyses that ultimately fail to replicate in randomized trials, regardless of how careful investigators are with the adjustment for other variables. This is because not all data that physicians use—sometimes subconsciously—in clinical practice are captured by standard variables and uncaptured data cannot be adjusted for. This is liable to be particularly relevant when analyzing SCT; because of the high intensity of the procedure, only the fittest patients will be recommended the procedure, and numerous considerations (medical, social, socioeconomic) are included in the evaluation of whether to recommend the procedure or not. A relevant cautionary tale here is mastectomy for metastatic breast cancer, a strategy that offered benefit to patients in carefully analyzed retrospective datasets, but did not in randomized trials [42,43,44,45]. Indeed, in the COMPLETE registry [39], despite the overall lack of significance in the multiple comparisons carried out, the difference in OS curves seemed greater than that of the PFS curves, perhaps suggesting that factors impacting OS beyond the disease itself were at play—similar to what was found for etoposide (see above). The lack of randomized data proving the benefit of autologous SCT in CR1 is particularly concerning because a growing number of experts and guidelines recommend this strategy [46,47] without a solid evidence basis, and such evidence will not be obtained if it becomes the standard of care on the basis of a (questionable) lack of equipoise. It should be noted that, thus far, consecutive clinical practice cohorts have shown little uptake of autologous SCT overall [21,38,39,48], even when considering only candidate patients that are young and in remission after induction, although this may reflect that the published cohorts contained data from many years back, given their mostly retrospective nature and the rarity of T cell lymphomas, requiring many years to reach informative sample sizes. Fortunately, a randomized clinical study from the LYSA group, TRANSCRIPT (NCT05444712), soon to start enrolment, will seek to determine the value of ASCT in CR1 by randomizing 204 patients with T cell lymphoma other than ALK + ALCL to autologous SCT or no consolidation in CR1 by PET.

For patients in partial remission, the benefit of autologous SCT has been far less studied, although the initial NLGT-01 trial [33] offered SCT to these patients. Indeed, very indirect evidence may indicate that patients with deeper remissions derive greater benefit from autologous SCT: (1) Data from the ECHELON-2 showed a substantially better PFS HR for autologous SCT in the BV arm (HR 0.36) than in the CHOP arm (HR 0.63) [36]; and (2) in another line of evidence altogether, a single center report found very high survival rates in patients in metabolic CR before autologous SCT [49]. However, this conclusion derives from indirect evidence, and it is unclear whether the conclusions would hold if high-quality evidence was obtained. Despite the lack of evidence supporting SCT in partial remission, a panel of transplantation experts for the American Society of Blood and Marrow Transplantation strongly supports autologous SCT for these patients [50] while the European Society for Blood and Marrow Transplantation does not provide guidance [46].

Regarding the choice between autologous and allogeneic SCT, a number of comparisons and one phase III clinical trial have compared them in CR1. Allogeneic SCT is well-known to provide long-term remissions in patients with PTCL, albeit with the usual caveat of a variable but often high non-relapse mortality [51,52,53,54]. The phase III trial from the French LYSA and the German Lymphoma Alliance [55] found no differences between the two types of consolidation in OS (3-yr OS probability 70% with autologous vs. 57% with allogeneic SCT) or PFS (3-yr EFS probabilities of 38% vs. 43%, respectively) in a trial where a large subset of patients did not reach transplant—confirming the suboptimal results of CHOP/CHOEP induction. Similar outcomes were seen despite a large difference in causes of death—progressive disease in the overwhelming majority of patients in the autologous SCT group vs. no relapse, but 31% non-relapse mortality in the allogeneic SCT (myeloablative conditioning). Retrospective studies largely confirm this idea although, again, the groups are rarely comparable because the type of SCT depends on the physician decision (rather than being randomized), so most patients in CR1 tend to receive autologous SCT and those that do not reach CR1 tend to be offered allogeneic SCT [54,56]. Regarding the specific aspects of allogeneic SCT, all donor sources seem to provide similar outcomes [57], and myeloablative conditioning does not seem to be beneficial over reduced-intensity conditioning [57,58]. A nuance that remains insufficiently explored is that different PTCL subtypes may be differentially sensitive to the graft-vs.-lymphoma effect, which appears greater in AITL [57,59].

A final comment with regard to rarer and highly aggressive forms of PTCL. Due to their extreme rarity, the evidence basis for therapeutic recommendations for enteropathy-associated and hepatosplenic T cell lymphomas is of very low quality. However, current consensus favors autologous SCT consolidation for the former, and allogeneic SCT for the latter [28,50].

To conclude, when should stem cell transplant be considered in the front-line? Although high-quality evidence supporting autologous SCT over no consolidation in CR1 is lacking, and there are large reports both supporting and not supporting autologous SCT, there seems to be a growing preference for the former over the latter. We think there is equipoise between the two and a randomized trial would be very informative. At present, an in-depth decision with the patient is warranted, aiming to let the patient know that this toxic procedure could very well not offer any benefit, but that relapsed PTCL has a poor prognosis and that obtaining a CR2 is not always possible.

Furthermore, should autologous or allogeneic SCT be preferred? For most patients in CR1, we would not offer allogeneic SCT given that there are no data indicating that it might be better than autologous SCT or no SCT. For patients with less than CR1 including PR or less when obtaining a better remission is not deemed possible, allogeneic SCT is the preferred option when aiming for a cure.

### 2.5. How Should Molecular Subtypes Impact Therapeutic Decisions?

In recent years, T cell lymphoma subtypes have been further subdivided based on the molecular findings. Most consistent among these are a split in ALK − ALCL and in PTCL, NOS. In 2014, two distinct subsets of ALK − ALCL, based on DUSP22 or TP63 rearrangements, were described [1,60]. The former, occurring in 30% of patients, was associated with distinct histopathological features (more frequent presence of hallmark cells and absence of cytotoxic markers) and a favorable prognosis (90% 5-yr OS probability), while the latter, occurring in 8% of patients, was associated with a poor prognosis (17% 5-yr OS probability), with the rest of the patients falling in between. The existence of these subsets, their prevalence, and their prognostic correlations was subsequently validated by the Danish group [61], and a focused characterization of DUSP22-rearranged ALCL confirmed them to be a biologically distinct subset [62]. A second validation was obtained from the British Columbia Cancer Agency (BCCA) cohort, although survival in all groups was poorer in this analysis (5-yr OS probability 40% for DUSP22-rearranged patients [63]). Additionally, while initially thought to be exclusive, rare isolated patients with coexisting DUSP22 and TP63 rearrangements have been reported [64], with histopathological findings and a prognosis that appears similar to those with TP63 rearrangements.

It should be noted that subdividing ALK − ALCL into subsets based on biological findings led to very small sample sizes (the original, Danish and BCCA cohorts included 22, five, and 12 patients with DUSP22 rearrangements, respectively), which led to very imprecise survival estimates, where one or two patients having a survival event may have a large impact on survival curves and change the *p* values from significant to non-significant.

Gene expression analyses of PTCL, NOS revealed that most patients could be grouped into either a GATA3 overexpression (~1/3 of patients) or a TBX21 overexpression subtype (~1/2 of patients) [65,66]. PTCL, NOS with overexpression of GATA3 are associated with a very poor prognosis [65,67,68]. PTCL, NOS with TBX21 overexpression are associated with a better—albeit not good—prognosis, although a subset of TBX21^high^ tumors have a cytotoxic gene expression profile and/or DNMT3A mutations and also seem to be associated with a very poor prognosis [65,69].

To conclude, how should patients with ALCL with DUSP22 or TP63 rearrangements be treated? It is clear that larger cohorts providing more reliable survival estimates for DUSP22-rearranged patients are needed before a different therapeutic approach from other PTCLs can be defended. However, the evidence supporting some therapeutic interventions in PTCL is already weak and DUSP22-rearrangement is likely to tip the scales for some patients and/or physicians. For instance, one might argue for CHOP alone, without etoposide or autologous SCT consolidation in CR1 for these patients, considering that the evidence for etoposide or autologous SCT consolidation in PTCL is quite weak. Conversely, for TP63-rearranged patients, while there was a rarer subset with very small sample sizes in all cohorts, treatment within promising clinical trials seems an obvious best course of action. Failing that, evidence for any specific therapeutic approach is lacking and we favor treating these patients with the preferred PTCL (other than ALK + ALCL) treatment at that institution.

Regarding the PTCL, NOS molecular subtypes, the prognostic difference between the groups seemed to be smaller, with both groups still associated with a fairly poor prognosis (even if poorer in GATA3^high^ tumors). Therefore, de-escalation of therapeutic intensity on the basis of molecular subtype does not seem to be warranted. Thus far, to our knowledge, no trials testing (or reporting on) subtype-specific therapeutic strategies have been conducted. In the future, however, with biology-adapted strategies, this is likely to change, with some authors postulating that GATA3^high^ PTCL, NOS might particularly benefit from phosphoinositide-3-kinase inhibition [70].

### 2.6. Is There a Role for Targeted Agents beyond BV?

#### 2.6.1. ALK Inhibitors

A number of ALK inhibitors are approved for ALK positive non-small cell lung cancer [71]. Given the acceptable side effect profile and the ostensible biological dependence of ALK + ALCL on the ALK rearrangement [72], their use for these patients is appealing. Despite this being a rare disease with an overall good prognosis, leaving little chance of testing these drugs in the relapse setting, some efforts have been published. Compassionate use of crizotinib was initially reported in nine patients with relapsed/refractory ALK + ALCL (unclear whether T or B cell lineage) of which four obtained prolonged responses and three were bridged to allogeneic SCT [73]. In a subsequent phase Ib trial including 17 patients with relapsed/refractory ALK + ALCL (again, with unclear lineage), an ORR of 53%, all but one of them CR, was reported [74]. Results in the adult population were seconded by a phase 2 trial in 20 children up to 20 years of age, with relapsed/refractory ALCL, 18 of which obtained a response with crizotinib [75]. Twelve of them were bridged to allogeneic stem cell transplant. This trial led to FDA approval of crizotinib for children and young adults with relapsed/refractory ALCL. Efforts to combine crizotinib with chemotherapy in the pediatric population are ongoing (NCT01606878, NCT01979536).

With alectinib, there is a phase 2 trial including 10 adult patients with relapsed/refractory ALCL of T cell lineage reporting an 80% ORR (60% CR) with five patients with ongoing (>1 year) responses and three patients bridged to allogeneic SCT [76]. This trial led to the approval of alectinib in Japan for this population. There have also been isolated case reports of patients with isolated CNS relapse of an ALK + ALCL successfully treated with alectinib [77].

Data with ceritinib and lorlatinib are scarcer, with one reported case of a young patient successfully treated with the former [78] and one with the latter [79]. There is an ongoing trial aiming to recruit and treat 12 patients with relapsed/refractory ALK + ALCL after at least one line of chemotherapy and one ALK inhibitor (NCT03505554).

To our knowledge, there are neither reports nor ongoing trials with ALK inhibitors in the front-line for ALK + ALCL.

#### 2.6.2. Histone Deacetylase (HDAC) Inhibitors

Histone deacetylase (HDAC) is an essential enzyme for the epigenetic regulation of gene expression. Preclinical and early clinical data have shown activity for HDAC inhibitors in T cell lymphomas and, indeed, romidepsin and belinostat are FDA-approved in the relapsed setting, in monotherapy, and largely based on single arm trials with response rates as primary endpoints [80,81,82]. Chidamide, another HDAC inhibitor, is approved by the Chinese FDA [83]. Of interest, response rates with HDAC inhibitors appear higher in patients with AITL and TFH phenotype lymphomas, in accordance with the greater frequency of mutations in epigenetic regulators (TET2, IDH, DNMT3, RHOA) seen in these subtypes.

These results ultimately led to a recent phase 3 trial of romidepsin-CHOP vs. CHOP for treatment-naïve PTCL. The trial was negative, with a median PFS of ~1 year in both arms [4]. Subgroup analyses did demonstrate a numerical advantage in PTCL with the TFH phenotype, but the trial was not powered to assess the benefit in specific subgroups. A phase 3 trial in the front-line setting randomizing patients to CHOP vs. CHOP plus azacytidine and chidamide (NCT05075460) is ongoing.

#### 2.6.3. Lenalidomide

Lenalidomide is an antineoplastic agent acting through a number of mechanisms [84] approved for the treatment of multiple B cell malignancies. However, very early trials have also shown potential activity in T cell lymphomas. A number of phase 2 trials in the relapsed setting [85,86,87] and two phase I trials in combination with romidepsin [88] confirmed this activity, particularly in AITL. This led to a phase 2 trial in the front-line setting in combination with CHOP where 78 patients with AITL (the diagnosis or AITL or PTCL with the TFH phenotype was only confirmed in 71) were treated [89]. The complete response rate and the 2-yr progression-free survival estimate were 41% and 42%, respectively, which did not appear to be better than the results with CHOP. A trial with the same regimen (although with a different lenalidomide schedule, NCT04423926), aiming to include 90 patients with any PTCL, not just AITL, is recruiting. Unsatisfactory results would likely end attempts to move lenalidomide into the front-line setting. At this time, evidence supporting the use of lenalidomide in PTCL is weak and only in monotherapy in the relapsed setting.

#### 2.6.4. Azacytidine

Azacytidine is a hypomethylating agent approved for the treatment of patients with myelodysplastic syndrome. Given the almost ubiquitous presence of mutations in epigenetic regulators seen in PTCL with the TFH phenotype, trials testing azacytidine in these patients seemed warranted. A phase I and small subsequent phase 2 trial in combination with romidepsin [90,91] and a retrospective experience in monotherapy in the relapsed setting [92] showed very promising response rates, up to 80% in PTCL with the TFH phenotype. However, a recent small phase 3 trial (NCT03593018), thus far published only in the abstract form, failed to show survival improvement with oral azacytidine over the investigator’s choice (mostly gemcitabine and bendamustine) in R/R AITL [93].

To conclude, is there a role for targeted agents beyond BV? At this time, there is no role for ALK inhibitors, histone deacetylase inhibitors, lenalidomide, or azacytidine in the front-line and their use remains restricted to patients in the relapsed setting.

## 3. Conclusions

Standard treatments for PTCL offer suboptimal outcomes and prognosis remains poor, even for fit patients. Given the low prevalence, evidence supporting any therapeutic intervention is often of low quality and a number of controversies remain vis-à-vis front-line treatment. We provide a focused discussion on these controversies. Overall, we found (1) no evidence supporting the use of CNS prophylaxis for most patients or any specific subgroup, although the high-risk in some ALK + ALCL cohorts makes further analysis imperative; (2) no clear evidence that etoposide provides a survival benefit for PTCL, other than for ALK + ALCL, where, although evidence is of low quality, is consistent, has a large effect size, and is without clear biases; (3) BV-CHP is the preferred induction regimen for ALCL other than ALK + ALCL with IPI0–2; (4) there is equipoise regarding SCT consolidation in CR1 and a clear recommendation cannot be provided for all patients, therefore, pending higher quality evidence, an in-depth discussion with the patient is warranted; (5) it is likely to be too soon to treat DUSP22 rearranged patients differently than other ALCL, but the omission of etoposide or autologous SCT consolidation may be considered, especially for physicians and patients that remain unconvinced of their value in ALCL and PTCL more generally; and (6) targeted agents such as ALK inhibitors, lenalidomide, azacytidine, or HDAC inhibitors should not be used in the front-line treatment of ALK + ALCL.

## Figures and Tables

**Table 1 cancers-15-00220-t001:** Selected studies comparing consolidation with autologous stem cell transplantation with other strategies in peripheral T cell lymphoma.

Author	Reference	Lymphoma Subtypes	Number of Patients	Induction Regimen	Consolidation Strategies	Survival in Patients in Complete Remission After Induction
Savage et al., 2022	36	ALCL, AITL, PTCL, NOS (mostly ALK − ALCL) in CR after induction	211	BV-CHP (*n* = 114) CHOP (*n* = 97)	Autologous SCT vs. no consolidation	BV-CHP + Auto SCT: 3-yr PFS 80.4% BV-CHP + no SCT: 3-yr PFS 54.9% CHOP + Auto SCT: 3-yr PFS 67.2% CHOP + no SCT: 3-yr PFS 54.1%
Advani et al., 2021	38	AITL	282	Anthracycline-based w/o etoposide 65%, anthracycline-based with etoposide 16% Other 19%	Autologous SCT vs. no consolidation	Auto SCT: 5-yr PFS 79% No auto SCT: 5-yr PFS 31% Auto SCT: 5-yr OS 89% No auto SCT: 5-yr OS 52%
Park et al., 2018	39	All PTCL	499	Anthracycline-based w/o etoposide 42%, anthracycline-based with etoposide 21% Other 37%	Autologous SCT vs. no consolidation	Auto SCT: 5-yr OS 87.8% No auto SCT: 5-yr OS 70.2%
Brink et al., 2022	3	ALK − ALCL, AITL, PTCL, NOS	213	CHOP or CHOEP	Autologous SCT vs. no consolidation	Auto SCT: 5-yr OS 82% No auto SCT: 5-yr OS 47%
Martin et al., 2022	37	ALK − ALCL, AITL, PTCL, NOS	174	CHOP (*n* = 126) CHOEP (*n* = 16) Other (*n* = 32)	Autologous SCT vs. no consolidation	Auto SCT: 5-yr PFS 63% No auto SCT: 5-yr PFS 49% Auto SCT: 5-yr OS 74% No auto SCT: 5-yr OS 62%
Janikova et al., 2019	20	All PTCL	906	Heterogeneous protocols	Autologous SCT vs. no consolidation	Auto SCT: 5-yr PFS 41% * No auto SCT: 5-yr PFS 46% * Auto SCT: 5-yr OS 49% * No auto SCT: 5-yr OS 59.5% *
Ellin et al., 2014	22	All PTCL	755	CHOP or CHOEP (*n* = 499)	Autologous SCT vs. no consolidation	Better for the auto SCT group (estimates not given) *
Schmitz et al., 2021	55	All PTCL other than ALK ALCL	104	CHOEP × 4 + DHAP × 1	Autologous SCT vs. allogeneic SCT (if donor available)	Auto SCT: 3-yr PFS 39% * Allo SCT: 3-yr PFS 43% * Auto SCT: 3-yr OS 70% * Allo SCT: 3-yr OS 57% *

* Compared groups were based on intention-to-transplant (or intent to treat in the Schmitz et al. trial) rather than based on the achievement of remission after induction. NB: All studies were retrospective other than that by Schmitz et al., 2022, which is a randomized clinical trial.

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
