# Peer review of "Controversies in the Front-Line Treatment of Systemic Peripheral T Cell Lymphomas"

_cancers, 2022, doi:10.3390/cancers15010220_

Round 1
Reviewer 1 Report
The lack of robust evidence base in PTCL management means that clinical practise varies significantly with most treatment protocols derived from historical dogma. Sorigue and Kuittinen address key controversies summarising available data.
Overall, the manuscript is clearly written. The controversies selected for discussion are highly relevant and of interest, apart from perhaps the section on ALK inhibitors given that this treatment approach is not widely available (and therefore limited controversy). The available evidence is succinctly summarised. The authors provide their own perspectives and conclusions – not everyone will agree given that these are contentious areas, but I feel most are balanced and appropriate.
I have the following specific comments:
Page 1
Line 34. It may be helpful to remind the reader that CHOP became the de facto standard of care following previous aggressive lymphoma trials that mostly comprised DLBCL.
Page 2
Line 88. The statement regarding lack of evidence of CNS prophylaxis should be softened and it would be more factually accurate to state that, whilst the data overall suggested no benefit, an advantage in specific high-risk subsets cannot be excluded given the lack of statistical power.
Page 4
Lines 159/159. Journal names don’t need to be included in the main text.
Line 164. I would rephrase the statement to say that subgroup analysis failed to show evidence of benefit in other subtypes but were not powered to draw robust conclusions
Line 171/172. The authors are right to highlight the fact that ECHELON-2 excluded ALK+ patients with an IPI 2 or less. However, there is no prospective randomised data that CHOEP is superior to CHOP for these patients either. Also, the IPI may inadequately identify higher risk ALK+ patients aged 40-60 years given that patients >40 years having been shown to have a worse outcome. Many have decided it is more appropriate to extrapolate from robust prospective randomised data rather than continue with either a de facto standard proven inferior in higher risk patients (CHOP) or use a regimen supported only by post-hoc or retrospective data (CHOEP).
Page 5
Line 208. Consider replacing aggressiveness with toxicity or intensity
Line 228. Not sure one can extrapolate this data to confirm differences in depth of remission.
Page 6
Line 258. Could mention the recently started LYSA led randomised study, TRANSCRIPT. This is now in the public domain https://www.haematologica.org/article/view/haematol.2022.280658/74259 . Or bring this into the debate page 5
Line 259. Suggest expanding the sentence and bringing in the point from lines 261/2.….At present, an in-depth decision with the patient is warranted…… covering potential toxicity, uncertainty in benefit, whilst also acknowledging the poor outcomes of relapsed PTCL.
Lines 261/2. I thought this was a bit ambiguous and it would be better to delete and finish at the previous sentence (but see comment above).
Page 8
Line 359. Should include RHOA in the recurrently mutated gene list. I would state may respond rather then seem.
Line 360. Suggest reference Ghione et al, Blood Advances 2020. Also correct spelling error for hypomethylating.
Line 362. “TFH lymphomas might have done better with the experimental arm” – this is a bit vague; consider rewording to be more accurate…..subgroup analysis demonstrated a numerical advantage in TFH lymphomas but statistical power was lacking to confirm significant benefit.
Line 364. Just to highlight that the ORACLE study oral abstract for ASH is now out – could consider updating the text to keep the article as current as possible. Failed to meet primary endpoint.
Line 367. AITL data on GVL should be included in the main text body rather than just brought into the discussion.
Author Response
Dear editor & reviewer,
Thank you for considering our manuscript for publication. And thank you for the careful and caring review of the manuscript. We also thank you for your encouraging words. We feel that, with your comments, a more precise description of findings and insights is now given in the manuscript
The point-by-point answers to your comments are below and the changes in the manuscript are highlighted in yellow.
The lack of robust evidence base in PTCL management means that clinical practise varies significantly with most treatment protocols derived from historical dogma. Sorigue and Kuittinen address key controversies summarising available data.
Overall, the manuscript is clearly written. The controversies selected for discussion are highly relevant and of interest, apart from perhaps the section on ALK inhibitors given that this treatment approach is not widely available (and therefore limited controversy). The available evidence is succinctly summarised. The authors provide their own perspectives and conclusions – not everyone will agree given that these are contentious areas, but I feel most are balanced and appropriate.
I have the following specific comments:
Page 1
Line 34. It may be helpful to remind the reader that CHOP became the de facto standard of care following previous aggressive lymphoma trials that mostly comprised DLBCL.
We have added a sentence to that effect.
Page 2
Line 88. The statement regarding lack of evidence of CNS prophylaxis should be softened and it would be more factually accurate to state that, whilst the data overall suggested no benefit, an advantage in specific high-risk subsets cannot be excluded given the lack of statistical power.
We have added a sentence adding nuance to our previous statement.
Page 4
Lines 159/159. Journal names don’t need to be included in the main text.
Deleted from the text
Line 164. I would rephrase the statement to say that subgroup analysis failed to show evidence of benefit in other subtypes but were not powered to draw robust conclusions
Rephrased
Line 171/172. The authors are right to highlight the fact that ECHELON-2 excluded ALK+ patients with an IPI 2 or less. However, there is no prospective randomised data that CHOEP is superior to CHOP for these patients either. Also, the IPI may inadequately identify higher risk ALK+ patients aged 40-60 years given that patients >40 years having been shown to have a worse outcome. Many have decided it is more appropriate to extrapolate from robust prospective randomised data rather than continue with either a de facto standard proven inferior in higher risk patients (CHOP) or use a regimen supported only by post-hoc or retrospective data (CHOEP).
We have added this consideration
Page 5
Line 208. Consider replacing aggressiveness with toxicity or intensity.
Done
Line 228. Not sure one can extrapolate this data to confirm differences in depth of remission.
Deleted
Page 6
Line 258. Could mention the recently started LYSA led randomised study, TRANSCRIPT. This is now in the public domain https://www.haematologica.org/article/view/haematol.2022.280658/74259 . Or bring this into the debate page 5.
Added
Line 259. Suggest expanding the sentence and bringing in the point from lines 261/2.….At present, an in-depth decision with the patient is warranted…… covering potential toxicity, uncertainty in benefit, whilst also acknowledging the poor outcomes of relapsed PTCL.
Rephrased
Lines 261/2. I thought this was a bit ambiguous and it would be better to delete and finish at the previous sentence (but see comment above).
Rephrased
Page 8
Line 359. Should include RHOA in the recurrently mutated gene list. I would state may respond rather then seem.
Done
Line 360. Suggest reference Ghione et al, Blood Advances 2020. Also correct spelling error for hypomethylating.
Done
Line 362. “TFH lymphomas might have done better with the experimental arm” – this is a bit vague; consider rewording to be more accurate…..subgroup analysis demonstrated a numerical advantage in TFH lymphomas but statistical power was lacking to confirm significant benefit.
Done
Line 364. Just to highlight that the ORACLE study oral abstract for ASH is now out – could consider updating the text to keep the article as current as possible. Failed to meet primary endpoint.
Done
Line 367. AITL data on GVL should be included in the main text body rather than just brought into the discussion.
Done
Thank you again for your careful review of the manuscript. We do appreciate it.
Reviewer 2 Report
Sorigue et al. gave a systematic review of controversies in front-line treatment of systemic peripheral T cell lymphomas based on clinical standpoints. But it seems to present the controversies or challenge in different aspects of treatment of PTCL. It’s better to add the clear advices based on the current evidence in each section. Please see the detail comments below.
Major comments:
1.It’s better to add the description of pathogenesis process of PTCL in the section of introduction. And give a brief introduction of controversies in treatment of PTCL based on the potential mechanism mediated reasons.
2. What about the gene characteristics of PTCL and the influence of gene features on the treatment, prognosis of PTCL in recent years?
3.Was there any predictive model for the risk of CNS relapse in PTCL? What methods/risk score were based to guide to CNS prophylaxis right now? What regimen were used for CNS prophylaxis in PTCL? High dosage MTX or intrathecal injection MTX?
4. It’s better to highlight the mortality and complications in older/ poor physical status patients limit the survival advantage of CHOEP in PTCL.
5.It’s was not clear of the role of Brentuximab in the treatment of PTCL. Was it really the first-line of treatment in PTCL?
6. What about others therapies in PTCL? Such as lenalidomide, HDAC inhibitors, PD1/PDL1 monoclonal antibody.
7. It’s better to description the future directions before conclusion in the manuscriptions.
8.It’s better list a Table for the comparison of auto hematopoietic stem cell transplantation consolidation versus other chemotherapies consolidation in PTCL. It’s better to give a figure of algorithm for the management/treatment of PTCL based on the current evidence.
Author Response
Dear editor & reviewer,
Thank you for considering our manuscript for publication. We appreciate the reviewer comments and think they have helped us improve the manuscript.
The point-by-point answers to your comments are below and the changes in the manuscript are highlighted in yellow.
Sorigue et al. gave a systematic review of controversies in front-line treatment of systemic peripheral T cell lymphomas based on clinical standpoints. But it seems to present the controversies or challenge in different aspects of treatment of PTCL. It’s better to add the clear advices based on the current evidence in each section. Please see the detail comments below.
Our manuscript focused on the controversies and, as such, it was not intended to provide practical advice. Other manuscripts in this same issue with focus on the treatment of front-line and relapse T cell lymphoma. Our manuscript is rather intended to help the more advanced reader find the areas where current recommendations rely on rather weak evidence and higher quality evidence should be sought going forward to provide a more solid foundation for therapeutic decisions. That said, we understand the reviewer’s concern and request for practical answers. We have referenced manuscripts that provide such answers in the introduction.
Major comments:
1.It’s better to add the description of pathogenesis process of PTCL in the section of introduction. And give a brief introduction of controversies in treatment of PTCL based on the potential mechanism mediated reasons.
We tried to briefly outline the pathobiology of T cell lymphoma but this cannot be done rigorously and as briefly as a manuscript that focuses only on clinical answers could allow. Therefore, we have provided a brief biological context in each section discussing drugs targeting specific pathways
- What about the gene characteristics of PTCL and the influence of gene features on the treatment, prognosis of PTCL in recent years?
We have added a discussion and potential impact on treatment of the GATA3 and TBX21 gene expression subtypes. This is in the section 5, which already dealt with influence of biological subtypes on treatment (previously only DUSP22 and TP63 ALCL)
3.Was there any predictive model for the risk of CNS relapse in PTCL? What methods/risk score were based to guide to CNS prophylaxis right now? What regimen were used for CNS prophylaxis in PTCL? High dosage MTX or intrathecal injection MTX?
All of this is now discussed in section 1. All evidence is taken from B cell lymphoma: More specifically: Predictive models, none validated; no risk score is broadly recommended to guide CNS prophylaxis now? CNS prophylaxis is with IV methotrexate or intrathecal prophylaxis but no evidence supports either or one better than the other.
- It’s better to highlight the mortality and complications in older/ poor physical status patients limit the survival advantage of CHOEP in PTCL.
Done
5.It’s was not clear of the role of Brentuximab in the treatment of PTCL. Was it really the first-line of treatment in PTCL?
Discussed in section 3. For ALK- ALCL, BV-CHP is the standard. For other PTCL, there is no consensus. Some argue that there is no evidence for BV-CHP over CHOP while others contend that BV-CHP is not really more toxic than CHOP and that ECHELON-2 included most PTCL so BV-CHP should be the standard for all.
- What about others therapies in PTCL? Such as lenalidomide, HDAC inhibitors, PD1/PDL1 monoclonal antibody
Previous section 6 on ALK inhibitors is now on all targeted agents, including lenalidomide, HDAC inhibitors and we have also added azacytidine. For PD1/PD-L1 targeted agents there is nothing (to our knowledge) in the front-line so we have not added it in this manuscript that centers on controversies in the front-line treatment of T cell lymphoma
- It’s better to description the future directions before conclusion in the manuscriptions
We initially made this change but with the addition of all data on targeted therapies (per comment number 6) all that we discussed in the “future directions” section was moved to section 6, so it was finally removed.
8.It’s better list a Table for the comparison of auto hematopoietic stem cell transplantation consolidation versus other chemotherapies consolidation in PTCL. It’s better to give a figure of algorithm for the management/treatment of PTCL based on the current evidence.
We have added a table of selected studies comparing auto SCT with other strategies. We have selected the large and recent relevant studies and referred the reader to a recent review in Cancers focusing specifically on Auto SCT in the front-line of T cell lymphoma. Regarding the figure, and in line with the comments above, we think it would undermine a manuscript precisely on the controversies of the front-line treatment. We have added recent references for the reader seeking the practical solutions to be able to find them expediently.
Reviewer 3 Report
The authors provide a comprehensive review of the current unresolved issues in the frontline treatment of PTCL.
Issues:
1. As regards the first question: role of CNS prophylaxis, the authors suggest that it could be considered in select situations but have not specified what these are.
Another issue which is unclear is whether the relapses are isolated to the CNS or synchronous with systemic relapses. The author's insight into this would be useful.
2. As regards the question of the addition of Etoposide to CHOP, the authors conclude that it is reasonable to consider this option in younger patients. It would be useful to know whether they recommend an age cutoff (?60 years) or a fitness/ tolerability assessment, as there does not appear to be an obvious difference in disease biology according to age at diagnosis.
3. The unusual stratification and enrolment plan of the ECHELON-2 study to restrict the non ALCL patients to 25% of the entire population means that the question around benefit of BV is difficult to answer. As currently available according to the label there is no distinction between various CD30 positive PTCL subgroups. It should be clearly stated that it is is author's opinion that it may not be of benefit in the non ALCL population as this recommendation is based on subgroup analysis which the study is not designed to answer.
Another important unresolved question is the threshold for CD30 positivity that should be used to consider the use of BV. Analysis of this issue would be useful.
4. In the section on the evidence for the role of SCT in PTCL, a mention of the role in rare subgroups such as EATL ( autologous) and hepatosplenic lymphoma ( allogeneic) would be useful as the limited evidence and consensus data suggest a benefit in these indications.
Author Response
Dear editor & reviewer,
Thank you for considering our manuscript for publication. And thank you for raising this unclear points and nuances. They helped us make the review clearer for the potential reader.
The point-by-point answers to your comments are below and the changes in the manuscript are highlighted in yellow.
1.As regards the first question: role of CNS prophylaxis, the authors suggest that it could be considered in select situations but have not specified what these are.
We have clarified that we think these situations exist (publications find subsets with high CNS relapse rates) but the evidence is still unclear on which subsets these are (because of inconsistencies between publications)
Another issue which is unclear is whether the relapses are isolated to the CNS or synchronous with systemic relapses. The author's insight into this would be useful.
Few series have analyzed this. We have found 3, 2 suggesting about 50% isolated-50% systemic relapse and one finding a 10% risk of isolated relapse. This is now in the manuscript. We think inconsistencies are likely due to both very imprecise estimates (the number of patients with CNS relapse in each study is very very small) but also potentially different restaging protocols (what are the indications for a lumbar puncture, some have a lower threshold to test, particularly if the patient will not be offered curative-intent treatment, which may be most patients upon relapse). More evidence needed.
- As regards the question of the addition of Etoposide to CHOP, the authors conclude that it is reasonable to consider this option in younger patients. It would be useful to know whether they recommend an age cutoff (?60 years) or a fitness/ tolerability assessment, as there does not appear to be an obvious difference in disease biology according to age at diagnosis.
At this time the evidence available is for an age cutoff, but fitness assessment is likely to be a better way to select candidates for more intensive treatment (evidence needed to support this). We have added this nuance to the text
- The unusual stratification and enrolment plan of the ECHELON-2 study to restrict the non ALCL patients to 25% of the entire population means that the question around benefit of BV is difficult to answer. As currently available according to the label there is no distinction between various CD30 positive PTCL subgroups. It should be clearly stated that it is is author's opinion that it may not be of benefit in the non ALCL population as this recommendation is based on subgroup analysis which the study is not designed to answer.
Indeed, it is our takaway from the study but do agree that others may reach other conclusions. This has been added to the text.
Another important unresolved question is the threshold for CD30 positivity that should be used to consider the use of BV. Analysis of this issue would be useful.
We concur that this is an important analysis going forward. We have added this to the manuscript.
- In the section on the evidence for the role of SCT in PTCL, a mention of the role in rare subgroups such as EATL ( autologous) and hepatosplenic lymphoma ( allogeneic) would be useful as the limited evidence and consensus data suggest a benefit in these indications.
Done
Thank you again for your constructive review of the manuscript